# The Effect of Interaction between Followers and Influencers on Intention to Follow Travel Recommendations from Influencers in Indonesia Based on Follower-Influencer Experience and Emotional Dimension

Betty Purwandari [1], Arief Ramadhan [2,*], Kongkiti Phusavat [3], Achmad Nizar Hidayanto [1], Adyssa Fairuz Husniyyah [1], Ferdinand Hanif Faozi [1], Nicolas Henry Wijaya [1] and Rifqi Hilman Saputra [1]

1   Faculty of Computer Science, Universitas Indonesia, Depok 16424, Indonesia
2   Computer Science Department, BINUS Graduate Program—Doctor of Computer Science, Bina Nusantara University, West Jakarta 11480, Indonesia
3   Faculty of Engineering, Kasetsart University, Bangkok 10903, Thailand
*   Correspondence: arief.ramadhan@binus.edu

**Abstract:** Social media has become a very commonplace way for many people to have social interactions. The role of social media has changed from what was originally only a way to bridge social interactions, to becoming a business tool in various industries, one of which is the tourism industry. The interaction between social media users can create new ways to increase public awareness of existing tourist objects. One way to achieve that goal is by utilizing social media influencers. This study aims to identify the factors that influence the intention of the followers to follow the travel recommendations given by the influencer. This study uses the theory of follower-influencer experience and the theory of emotional dimensions, as well as their effect on the level of commitment and intention to follow the recommendation. This research was conducted by distributing surveys through social media and we managed to obtain a total of 203 valid respondents. The results of the study were analyzed using structural equation modeling (SEM), which showed that information experience and homophily experience had a significant effect on pleasure, arousal, and dominance. Pleasure and dominance have a significant effect on commitment, and commitment has a significant effect on the intention to follow the recommendation.

**Keywords:** travel recommendations; influencers; followers; social media; follower-influencer experience; emotional dimensions; commitment; intention to follow recommendations

## 1. Introduction

Currently, there are more than 3.5 billion people who use social media. Various social media applications are used, such as Instagram, Facebook, and Twitter. Through the application, they can connect with other social media users [1]. Social media can also be used to make calls, send messages, and create and share content [2].

At the beginning of its emergence, social media was often used for entertainment purposes only. However, nowadays the use of social media is not just for entertainment. Several social media channels are also commonly used for business purposes, especially marketing. One of the business activities that utilize the use of social media is business activities in the tourism sector that use social media as a marketing tool. According to a survey conducted by Stackla (www.stackla.com accessed on 12 June 2022), a visual content engine provider company, marketing activities using social media have a significant impact on tourist attractions. This is shown by the results of a survey which states that 86 percent of people feel interested and 52 percent of people make plans to go to a tourist spot after seeing photos or videos of the tourist spot on social media [3].

One of the main reasons for the success of tourism marketing activities on social media is influencers [4]. In a marketing activity, influencers act as parties who convey messages from the company concerned to their followers by relying on the previously built follower-influencer relationship [5]. Several previous studies have discussed the influence of influencers in marketing activities for tourist attractions on social media. One of them is the research conducted by Ong and Ito [6] which states that influencers are the main key to the marketing strategy of tourist attractions on social media because they are able to change the behavior of followers and their perception of tourist destinations so that followers tend to think about the tourist spot. Another study conducted by Shuqair and Cragg [7] also states that influencers play a big role in marketing tourist attractions on social media because influencers can change the perspective that followers have about a tourist place.

The relationship between the follower and the influencer can greatly influence the follower's decision to take certain actions. Brands usually collaborate with influencers to promote their products to their target customers [8]. The influencers have acted as brand ambassadors on social networking sites [8]. In our study, the influencers do not promote a brand but raise information about tourism locations that are worthy of being traveled to. Those lifestyle influencers share their personal opinions and experiences about the tourism location. Their posts can shape consumers' (followers') expectations [9].

The relationship between followers and influencers is strongly influenced by the influencer's posting habits. For example, if he is used to posting about food and then is suddenly promoting a car product, it tends to provide a negative experience for his followers. On the other hand, if influencers are accustomed to posting healthy lifestyles and then promoting healthy drinks, they tend to provide positive experiences for their followers. This is consistent with the results of Belanche et al. [8], which show that the consistency of the influencer affects the assessment from the follower of the influencer.

We propose that the relationship between influencers and followers will provide a certain experience to followers so that it affects the emotions of followers which in the end also affect the intention of followers regarding whether to follow the advice of the influencer or not. We called that experience an influencer-follower experience. In our study, we state that followers are customers who will "buy" suggestions from influencers to travel to the tourism spots suggested by influencers. The literature defines customer experience as "a multidimensional construct focusing on a customer's cognitive, emotional, behavioral, sensorial, and social responses to a firm's offerings during the entire purchase journey" [10]. We will adopt several theories of experience in our proposed model.

We adopt the information experience, entertainment experience, homophily experience, and relationship experience from Wang et al. [11]. We adopt those four experiences in our study because they are suitable for the context of the social media relationship between followers and influencers. Information experience is about the experience of value that comes from the follower's perception of the usefulness of information from influencers [11]. Adopting Nambisan and Watt [12], we can state that the entertainment experience is about the pleasant experience that customers (followers) have when they interact with the community members (influencers and other followers). The homophily experience is related to the similarity between the followers and influencers. Similarity can be in the form of personal traits such as age, gender, education [13], or political ideology [14]. The strong relationships experienced between influencers and their followers can promote the followers' need to purchase/follow what the influencer uses/ is doing [15].

In addition to the experience between followers and influencers, we suggest that the emotional dimension can also be a mediator that influences follower decisions. We adopt the pleasure-arousal-dominance (PAD) model to evaluate complex emotions from Mehrabian [16,17]. The model is considered to be one of the most influential models in the research about emotion [18]. Pleasure is related to positive or negative reactions of people to their environment [19]. Pleasure is the degree of joy or happiness of a person in a situation [20]. Arousal is about the motivation caused by the existing environment [21].

Arousal is also related to excitement [20,22]. During arousal, a person will feel enthusiastic, alert, and active [23]. Based on Russell and Mehrabian [24], we can define dominance as the degree to which a person feels powerful regarding their environment. If he can control or influence the situation surrounding him, it makes him feel dominant [19]. Several studies have proven that the PAD has an influence on user intention to use or customer intention to purchase some goods [19,25–29]. The model can also be used to capture online behavior [30]. Therefore, we state that this PAD will also affect online social media followers' intention to follow the recommendation from influencers. The PAD model is also one of the most popular approaches to studying tourist behavior and experience [31], and that is in line with the context of our study regarding the travel recommendation. Thus, we will adopt the PAD in this study.

We also suggest that commitment can be the mediator dimension between the emotional dimension and the follower intention. We draw this suggestion based on Wang et al. [11], who revealed that commitment can be interpreted as an emotional bond between the customer and the company or brand. That emotional bond can also appear between the follower and influencer, so we add the commitment dimension to our proposed model.

Based on the explanation above, this study combines follower-influencer experience with emotional dimensions and commitment in one model. To the best of our knowledge, this is the first time those three things have been combined. This provides an academic contribution to knowledge, especially in terms of followers' intention to follow recommendations from influencers. We will focus our discussion on those dimensions further in the context of travel recommendations. This will provide a practical contribution, especially for tourism or traveling companies, as to the importance of influencers to build and maintain their business. The results of this study can give insight to the companies about how they can provide the best experience, provide positive emotion, and increase follower commitment. To carry out the research, we will collect data through a survey. The results of the data that have been collected will be processed and analyzed using the structural equation modeling (SEM) technique with the partial least square (PLS) algorithm through the SmartPLS 3 application from SmartPLS GmbH, Oststeinbek, Germany

## 2. Literature Review

### 2.1. Social Media

Social media is one of the modern communication media outlets based on internet services. Some examples of social media referred to in this article are Facebook, YouTube, Instagram, TikTok, and various other social media. Social media gives us the ability to share information more easily, open up the world to be wider, and connect with more people in the world [32]. Social media is a place that can connect two or more people who are very far away, allowing them to communicate as if they were in the same place. Almost all aspects of life are affected by the existence of social media, from the economy, government, and health, to even education [32]. Social media can be used as a product or brand promotion media. Therefore, it is important for marketers to understand how to use social media to have a positive impact on their brands [5].

Currently, social media has many important roles in tourism development [33]. If previously the search for destinations was mostly focused on using search engines, now social media has become a new way for tourists to search for and find tourist attractions or destinations [33]. Tourists have used social media to help them in their decision-making process [34]. In addition to tourists, social media is also useful for tourism business actors for their business management [34,35]. It can help the company to create the proper marketing strategy [35,36]. Social media has enabled tourism-related companies to deal directly with their customers through the internet without physical engagement [35]. The company can use social media as a tool to analyze the tourists' attitude [37], preferences, and behavior. The results of the analysis can help tourism-related companies provide "a unique value proposition" to their customers [37].

## 2.2. Social Media Influencer

A social media influencer is a person, group, or brand that has the power to influence the buying habits of others by uploading some form of original or sponsored content to their social media platforms. The presence of social media influencers is a new form of marketing in social media. Someone who becomes a social media influencer is characterized by the number of followers on their social media accounts; almost everything they upload on social media receives a lot of attention and engagement, such as likes, comments, and so on. The presence of social media influencers can assist in marketing activities by influencing potential customers to buy a product or brand [38]. The honesty and openness of influencers, through the content they create, can have high influence and credibility in the eyes of the public. Campbell et al. [39] add that by using the help of social media influencers, a brand can achieve strategic goals and create a more positive customer experience.

Živković et al. [37] said that tourists currently trust the opinions of other travelers' on social media rather than marketing advice from official sources. Thus, an influencer who is also a traveler has the potential to have a big influence on his followers. Travel influencers can become opinion leaders in social media [40]. Travel influencers can visit the tourism destinations and promote that location and their experience of the destination [41]. Nowadays, travel companies have realized the potentiality of travel influencers [40] and they use them as a marketing tool. The existence of influencers has given rise to a new marketing practice called influencer marketing. Femenia-Serra and Gretzel [42] suggest that tourism destination management organizations have to incorporate influencer marketing in their long-term planning and strategic vision.

## 2.3. Follower-Influencer Experience

The follower-influencer experience is an experience that occurs between followers and influencers. In this study, influencers are people who have the influence to spread information to their followers through social media. The construct components of the follower-influencer experience in this study are adopted from Wang et al. [11], i.e., information experience, entertainment experience, homophile experience, and relationship experience. All of those follower-influencer experience definitions have been explained in our Introduction section. Influencers who have built a sizeable network of social media followers have the potential to exert influence on their followers [42]. The more followers an influencer has, the higher the probability that the followers of the influencer will see something uploaded by the influencer. Research conducted by Dhanesh and Duthler [43] shows that the relationship between followers and influencers has a positive influence on purchase intention. In the context of tourism, the relationship of influencers with their followers has been considered to be very valuable for destination management operators [41]. The tourism operators and destination marketers have to select appropriate influencers that can give a positive experience to their followers [44]. Influencers can build social (para-social) relationships with followers using responses and comments [45]. It has been revealed that followers of an influencer have a better positive relationship of intention to visit tourist destinations and a more positive attitude than non-followers [44].

## 2.4. Emotional Dimension

The emotional dimension is a concept used to describe the emotional state of a person [17]. The PAD model from Mehrabian [17] can be used to capture the individuals' emotional states. It consists of pleasure, arousal, and dominance [46]. The three aspects of the PAD model are considered capable of helping researchers to know a person's emotional condition more simply [17]. The first aspect is pleasure, which is an emotional dimension where a person has a very unhappy (sad) mood and vice versa. The second is the arousal aspect, which is a person's emotional dimension, which is closely related to passion, lust, and a sense of being motivated. Finally, the dominance aspect is a person's emotional dimension, where a person feels he has control or influence over something, and a person does not feel controlled or influenced by it [46].

Several studies have been conducted to determine the effect of pleasure, arousal, and dominant emotion of tourists on tourist destinations. Bigne [47] analyzed the effect of pleasure and arousal for visitors to a theme park on perceived satisfaction and their behavioral intentions. Ortiz-Ramirez et al. [48] found that pleasure, arousal, and dominance influence the behavior of pedestrians on sidewalks in Bogota, Colombia. The positive effect of entertainment and the design of a re-enactment of arousal and pleasure have been proven by [49]. However, it was found that only pleasure had a significant direct impact on satisfaction [49], while arousal only had a significant indirect effect through the mediation of pleasure on satisfaction [49]. Song et al. [50] evaluated the effect of entertainment experience, educational experience, escape experience, and esthetic experience on pleasure, arousal, and dominance toward satisfaction of theme-event events. It was found that entertainment experience, educational experience, and esthetic experience have a positive influence on arousal and pleasure [50]. In addition, only educational experience and esthetic experience have a positive influence on dominance [50].

*2.5. Commitment*

Commitment is a key feature needed by a company or brand to maintain a fruitful relationship with customers. Commitment can also be interpreted as a strong and emotional bond that is felt by a customer to a company or brand. In an online community, commitment can be shown by content creation behavior such as adding comments and liking a post. Various studies state that commitment to an online community does not exist by itself but is influenced by various other factors [11]. According to Kuo and Feng [51], the interaction characteristics of the company with customers and the benefits perceived by customers have a positive influence on commitment. Meanwhile, research conducted by Hur et al. [52] shows that commitment is significantly influenced by customer trust in an online community. Several research studies have been conducted about commitment to the tourism industry. Moghavvemi et al. [53] stated that community commitment can support the development of tourism. Meanwhile, Zheng [54] showed that positive emotions have a large impact on the commitment of residents toward tourism performing arts.

*2.6. Intention to Follow Recommendation*

Intention to follow a recommendation, also known as purchase intention, is a variable that can be measured by three behaviors, i.e., recommending, buying, or thinking about buying [11]. According to motivation theory, the intention to follow recommendations can be increased by customer satisfaction. The improvement process begins with customer satisfaction, which will cultivate customer commitment to the company. Commitment can increase the intensity of customer participation in the corporate community so that the intention to follow recommendations from customers will also increase [55]. This theory has been proven by a study that discusses the identification factor of the user brand community in regards to Facebook. The results of that study indicate that the commitment and relationship status of a company's community can be converted into a marketing effect, one of which is the intention to follow recommendations [56]. We suggest those types of intention to follow also occur when followers see travel recommendations given by influencers.

There are several studies that have been successful in analyzing the factors that influence the intention to follow recommendations. Jimenez-Castillo and Sanchez-Fernandez [57] revealed that perceived influence, brand engagement in self-concept, and brand expected value are in line with the intention to follow recommendations. According to Demiray and Burnaz [56], the intention to follow recommendations is influenced by the brand commitment and brand relationship factors. Ki et al. [42] found that social media influencers, influencer marketing, influencer persona, and human brands are in line with the intention to follow recommendation. Balakrishnan et al. [58] stated that electronic online word of mouth (E-WOM), online communities, and online advertisements are based on the intention to follow recommendations.

## 3. Materials and Methods

### 3.1. Research Model

In the Introduction section, we have explained that we will use follower-influencer experience, emotional dimensions, and commitment in our research model. The reasons for choosing the three dimensions have also been presented in the Introduction section. The combination of the three dimensions is the contribution of this study because there are no other studies that combine those three things. Our model is used to ascertain whether the experience between followers and influencers can affect emotional dimensions, i.e., pleasure, arousal, and dominance, and their effect on commitment and intention to follow recommendations. The research model used can be seen in Figure 1.

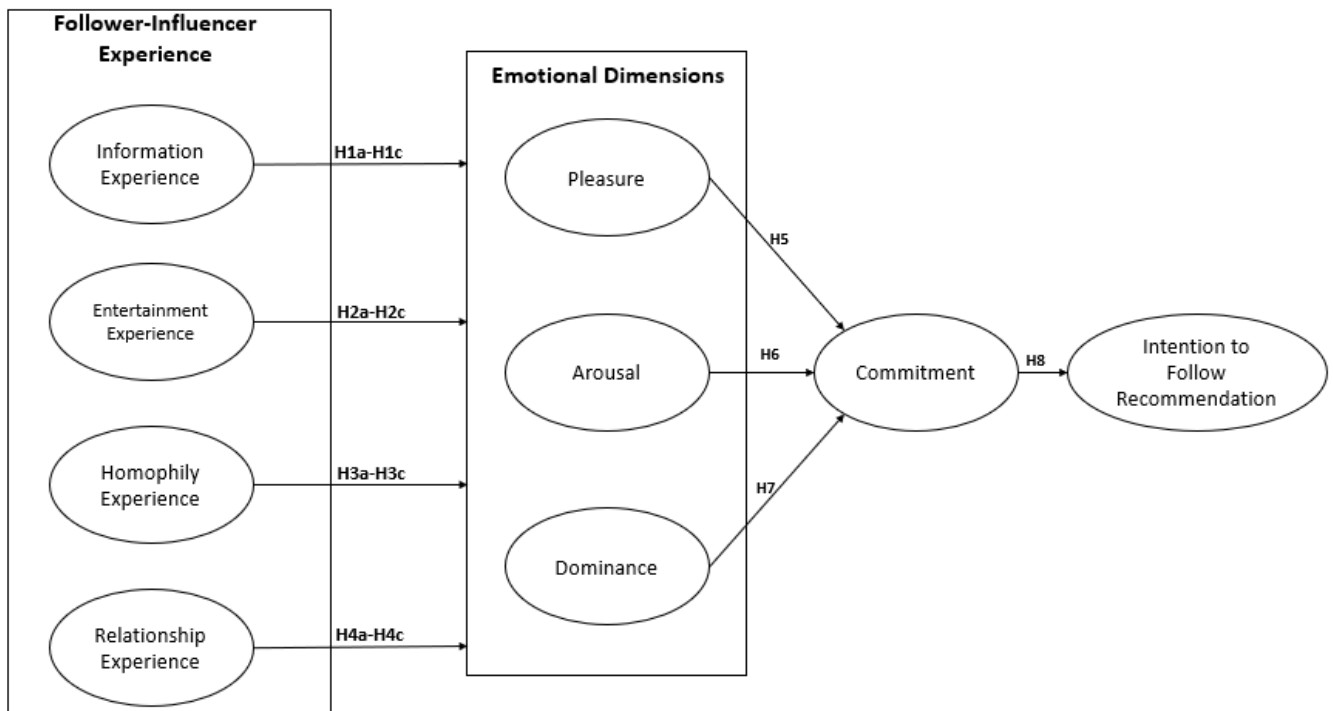

**Figure 1.** Research model.

### 3.2. Research Hypotheses Formulation

3.2.1. Information Experience and Pleasure, Arousal, and Dominance

According to Maybee et al. [59], information experience is a domain that discusses how a person obtains information and the meaning of information from various aspects of their lives. Information experience in the form of tourism literacy can provide new and different experiences [60]. Research finds that if someone receives tourism information from someone else or the tourism community on a platform, then the information can provide a useful experience, provide satisfaction for the recipient of the information, and increase interest and commitment to the tourism object [60]. The positive information experience obtained by followers from travel recommendations given by influencers will have a positive impact on increasing the pleasure, arousal, and dominance felt by followers. The reason is that followers feel that they have a good information experience, so they feel that the travel recommendations given by influencers are quite informative. Although followers do not necessarily follow the travel recommendations given, an informative impression can add new insights for followers that may be useful. In addition, with this informative impression, followers attain convenience, because they feel it is not difficult to search for tourist information that has been recommended before. Based on this explanation, the following hypothesis is formulated:

**H1a.** *Information Experience has a positive effect on Pleasure.*

**H1b.** *Information Experience has a positive effect on Arousal.*

**H1c.** *Information Experience has a positive effect on Dominance.*

3.2.2. Entertainment Experience and Pleasure, Arousal, and Dominance

Entertainment experience is everything related to triggering an increase in affection, physical indications, and moral motivation [61]. An individual (on a platform) has a tendency to find pleasure in himself by interacting with other people [62]. For example, someone follows an influencer's account on the Instagram platform, to find entertainment in the content presented by that influencer [62]. The comfort that followers obtain or feel about travel recommendations that influencers provide will have a good impact on increasing the pleasure, arousal, and dominance felt by followers. The justification for the previous statement is that if the follower of the influencer is entertained, he will feel that the influencer can bring joy to him. Based on this explanation, the following hypothesis is formulated:

**H2a.** *Entertainment Experience has a positive effect on Pleasure.*

**H2b.** *Entertainment Experience has a positive effect on Arousal.*

**H2c.** *Entertainment Experience has a positive effect on Dominance.*

3.2.3. Homophily Experience and Pleasure, Arousal, and Dominance

Adopted from the paper by Lawrence and Shah [63], homophily experience is a basic pattern in human relationships in the form of a person's tendency to equate preferences or treatment with others. This concept is related to social identity theory, which states that if a person has the same preference for an individual or reference group, a person will feel a positive emotional connection and perceive those around him as being "in the same frequency" as him [64]. The homophily experience felt by followers towards influencers who provide travel recommendations will have a good impact on increasing the pleasure, arousal, and dominance felt by followers because followers feel that there are similarities with the influencers in the perspective on a certain aspect. Based on this explanation, the following hypothesis is formulated:

**H3a.** *Homophily Experience has a positive effect on Pleasure.*

**H3b.** *Homophily Experience has a positive effect on Arousal.*

**H3c.** *Homophily Experience has a positive effect on Dominance.*

3.2.4. Relationship Experience and Pleasure, Arousal, and Dominance

According to Wang et al. [11], relationship experience is one of the factors that have a major impact on community commitment. The source also added that communities on social media, such as Facebook, can make it easier for people who are friends in the real world to connect online and also make it easier to make new friends. Ross et al. [65] added that Facebook is moving people's real-world relationships online. In addition, Habibi et al. [66] said that through the function of social networks and activities, people have the opportunity to get to know each other. Relationships that are well established between followers and influencers who provide travel recommendations have the potential to increase the pleasure, arousal, and dominance of followers. Based on this explanation, the following hypothesis is formulated:

**H4a.** *Relationship Experience has a positive effect on Pleasure.*

**H4b.** *Relationship Experience has a positive effect on Arousal.*

**H4c.** *Relationship Experience has a positive effect on Dominance.*

### 3.2.5. Pleasure, Arousal, Dominance, and Commitment

Pleasure-arousal-dominance is a model devised by Mehrabian and Russell [21]. Pleasure is a condition that can cause pleasant or unpleasant emotions to the user. Arousal is the level of intensity of pleasant or unpleasant emotion. Finally, dominance is the extent to which a person can control an event rather than being controlled by an event [21]. In conclusion, pleasure, arousal, and dominance are the three basic dimensions that indicate the state of feelings that humans have [67]. Research conducted by Reydet and Carsana [68] shows that the positive influence of human emotions, namely pleasure, arousal, and dominance, has a positive influence on commitment. Thus, with the increase in pleasure, arousal, and dominance felt by followers, there is expected to be an increase in commitment to the influencers they follow. Based on this explanation, the following hypothesis is formulated:

**H5.** *Pleasure has a positive effect on Commitment.*

**H6.** *Arousal has a positive effect on Commitment.*

**H7.** *Dominance has a positive effect on Commitment.*

### 3.2.6. Commitment and Intention to Follow Recommendation

Commitment is a strong and emotional bond that a person feels towards something. Various studies state that commitment is one of the most important factors in building intention to follow recommendations [11]. One reason is that commitment can encourage customers to support all products and brands related to the company [69]. In addition, commitment can also make the customer's view of the company more positive so that customers tend to buy the company's products [70]. This statement is proven by Demiray and Burnaz [56] in their research which shows that commitment can be converted into a marketing effect, one of which is the intention to follow recommendations. In addition, research conducted by Shi et al. [55] also proves that commitment can increase the intensity of customer participation in the corporate community so that the intention to follow recommendations from customers will also increase [55]. Based on the above explanations, we suggest that the commitment from followers to their influencers will have a positive effect on the intention of followers to follow recommendations from influencers, so we formulate the following hypothesis:

**H8.** *Commitment has a positive effect on Intention to Follow Recommendation.*

### 3.3. Research Instrument

The research instrument was made by adopting previous studies related to the dimensions used in this study. Then, the readability and interpretation test was carried out on each research instrument that had been made. The purpose of the readability test is to find out how well the respondents understand the instruments that have been made so that later the questionnaire data become accurate and can be used. The readability test was assisted by an expert and five people who were included among the target respondents. The selected expert has research experience in the topic of Information Systems and Information Technology. Several revisions to the statements on instruments have been made to ensure they are understandable. The instruments used for this research can be seen in Table 1.

**Table 1.** Research instrument.

| No. | Variable | Operational Definition | Question | Adopted from |
|---|---|---|---|---|
| 1 | Information Experience | Information about a tourist attraction that followers obtain from influencers | The influencers I follow are well-known as a source of information on tourist attractions. | [11] |
| | | | The influencers I follow provide accurate information about tourist attractions | |
| | | | Travel information from influencers I follow is useful to me. | |
| 2 | Entertainment Experience | Everything is related to triggering an increase in followers' affection due to travel recommendations given by influencers | My experience following the influencers has been great. | [11] |
| | | | My experience following the influencers adds to my happiness. | |
| | | | My experience following the influencers has been very encouraging. | |
| 3 | Homophily Experience | The similarity of travel preferences between influencers and followers' interests. | Travel experiences shared by influencers match my travel style. | [11] |
| | | | The influencers I follow have a travel style similar to mine. | |
| | | | The influencers I follow have the same travel tendencies as I do. | |
| 4 | Relationship Experience | The relationship experience between followers and influencers is related to travel recommendations given by influencers | People I know also follow the influencers I follow. | [11] |
| | | | I follow influencers because of recommendations from friends. | |
| | | | I recommend to people I know to follow the influencer accounts I follow. | |
| 5 | Pleasure | The extent to which followers are happy with the travel recommendations given by influencers | I am pleased after following influencers who provide travel recommendations. | [46] |
| | | | I feel happy after following influencers who provide travel recommendations. | |
| | | | I feel satisfied after following influencers who provide travel recommendations. | |
| 6 | Arousal | The level of emotional intensity that arises from followers when influencers provide travel recommendations | I feel energized after following influencers who provide travel recommendations. | [46] |
| | | | I feel passionate about traveling after following influencers who provide travel recommendations. | |
| | | | I feel enthusiastic about traveling after following influencers who provide travel recommendations. | |
| 7 | Dominance | The extent to which followers can control themselves when influencers provide travel recommendations | I can control what travel information I will receive from influencers. | [29] |
| | | | I feel that I can control the experience I will get from influencers. | |
| | | | When I follow these influencers, I can control the travel recommendations I will get. | |
| | | | When I follow this influencer, I can choose what travel information I want to see. | |

**Table 1.** *Cont.*

| No. | Variable | Operational Definition | Question | Adopted from |
|---|---|---|---|---|
| 8 | Commitment | How far do followers want to continue to follow and hear travel recommendations from influencers? | I have a sense of belonging to influencers. | [11] |
| | | | I will continue to follow social media influencers who provide travel recommendations. | |
| | | | I exchange information about tourist attractions with influencers and followers of influencers. | |
| | | | I will seek travel recommendations from influencers. | |
| 9 | Intention to Follow Recommendation | Followers' intention to follow travel recommendations from influencers | I will pass on the travel recommendations from influencers to others. | [11] |
| | | | I will visit tourist attractions recommended by influencers. | |
| | | | There is a possibility that I will consider visiting tourist attractions recommended by influencers. | |

After passing the readability test stage, the instruments were distributed online through social media in the form of a questionnaire. The questionnaire was divided into three parts: the first part is in the form of validation questions related to the research topic, the second part is in the form of demographic information of respondents, and the third part contains questions in the form of measurements for each research instrument using five-point Likert scales. Respondents were asked to give an assessment in the form of a value of 1 to 5, with the provision of a value of 1 indicating strongly disagree, 2 indicating disagree, 3 indicating neutral, 4 indicating agree, and 5 indicating strongly agree. The criteria to become respondents in this study are people who use social media and who follow influencers who recommend tourism on their social media.

*3.4. Data Collection*

The method used to collect data in this research is a survey using a questionnaire. The questions in the questionnaire were compiled based on the abovementioned research instruments. Questionnaires were distributed randomly and widely to the target respondents. The target respondents for our research are Indonesian citizens that have become followers of influencers who provide travel recommendations on social media. We leave it entirely to the respondents, as to what kind of account is considered an influencer in the tourism sector. However, we only take data from respondents who have followed their influencers for at least one year. Thus, we assume that the follower has known for a long time that the account being followed is actually acting like an influencer. We also explained to respondents that what is considered a tourism influencer is if the influencer often posts items about tourism in a well-known tourism area (e.g., Labuan Bajo, Bali), recommendations for natural tourism objects (e.g., mangrove forests, national parks), recommendations for building tourist objects (e.g., museums, historic buildings), as well as recommendations for experiencing tourism (e.g., paragliding activities, skateboarding).

The questionnaire was created using Google Form, and the link information was distributed on several social media channels, such as LINE, Instagram, Twitter, WhatsApp, and several other social media. The data collection was carried out from 6 May 2021, to 27 June 2021. We managed to obtain 238 respondents, 34 of whom were considered not to follow an influencer, so the data were considered invalid. The rest are considered valid data because we see that the respondents really follow influencers, so we obtain valid data from 203 respondents.

### 3.5. Data Analysis

After collecting data, the next step is to process and analyze the data. The purpose of this stage is to obtain indicators that represent the influence of follower-influencer interactions on the intention to follow travel recommendations from influencers based on the perspective of follower-influencer experience and emotional dimensions. Data processing and analysis were carried out using the structural equation modeling (SEM) technique, which is a technique that integrates various statistical analysis methods such as multiple regression, factor analysis, and path analysis [71]. SEM technique consists of two types of models, i.e., the measurement model and the structural model. The measurement model is used to test the latent and composite variables, while the structural model is used to test the dependencies of all hypotheses based on the results of path analysis [72].

In this study, we discuss the causal relationship between follower-influencer interactions and the intention to follow travel recommendations from influencers. The SEM technique algorithm that is suitable for explaining causal events as discussed in this study is the partial least square (PLS) algorithm. Therefore, data processing and analysis in this study were carried out using the PLS algorithm. In addition, the characteristics of the collected data are also suitable for processing using the PLS algorithm. The characteristics of our data fall into PLS-wise data characteristics, which are a relatively small size (less than 300), not normally distributed, and varied. The use of the PLS-SEM technique to process and analyze data is carried out by the SmartPLS 3application from SmartPLS GmbH, Oststeinbek, Germany.

## 4. Results

### 4.1. Respondents' Demographics

To find out the characteristics of the respondents, we collected data about gender, last education level, age, domicile, and social media used by respondents. We also asked how frequently they search for tourist recommendations from the influencers. Of the 203 data samples collected, there are 60.6% of female respondents and 39.4% of male respondents. The majority of respondents have an age range of 17 to 25 years, which is 83.3%. In addition, there are 1.5% of respondents aged less than 17 years, 10.3% aged 26–35 years, 2.5% aged 36–45 years, and 2.5% aged over 45 years. Most of the respondents live on Java Island, i.e., 87.7% of respondents are from Greater Jakarta and 9.4% are from outside Greater Jakarta. From outside Java Island, 2% of respondents live on Sumatra Island and 0.5% live on Kalimantan Island. Based on the last education level taken, respondents can be divided into four groups, i.e., bachelor (77.3%), senior high school (12.8%), diploma (5.9%), and master (3.9 percent).

The most widely used social media by respondents is Instagram, which reaches 97.5% of users. Below Instagram, there is YouTube with 80.3% of users. In addition, 60.6% of the respondents use Twitter, 30.5% use Facebook, and 9.9% use TikTok. A small percentage of respondents also use WhatsApp, Line, Pinterest, and Clubhouse. The demographic data of respondents in more detail can be seen in Table 2.

**Table 2.** Respondent demographic data.

| Gender | # | Percentage |
|---|---|---|
| Male | 80 | 39.4% |
| Female | 123 | 60.0% |
| **Age** | **#** | **Percentage** |
| <17 | 3 | 1.5% |
| 17–25 | 169 | 83.3% |
| 26–35 | 21 | 10.3% |
| 36–45 | 5 | 2.5% |
| >45 | 5 | 2.5% |
| **Domicile** | **#** | **Percentage** |
| Java Island (Greater Jakarta) | 178 | 87.7% |
| Java Island (Outside Greater Jakarta) | 19 | 9.4% |
| Sumatera Island | 4 | 2% |
| Kalimantan Island | 1 | 0.5% |
| **Education** | **#** | **Percentage** |
| Senior High School | 26 | 12.8% |
| Diploma | 12 | 5.9% |
| Bachelor | 157 | 77.3% |
| Master | 8 | 3.9% |
| **Social Media Used** | **#** | **Percentage** |
| Instagram | 198 | 97.5% |
| Twitter | 123 | 60.6% |
| Facebook | 62 | 30.5% |
| YouTube | 163 | 80.3% |
| TikTok | 20 | 9.9% |
| WhatsApp | 2 | 1% |
| Line | 1 | 0.5% |
| Pinterest | 1 | 0.5% |
| Clubhouse | 1 | 0.5% |

Based on the data that have been collected and summarized in Table 3, it can be seen that as many as 79 of the 203 respondents searched for tourist recommendations from influencers 2 to 4 times. Meanwhile, 78 other respondents sought recommendations less than 2 times for tourist attractions from influencers.

**Table 3.** Respondents' intensity in seeking travel recommendations from influencers.

| Intensity (Times) | # | Percentage |
|---|---|---|
| <2 | 78 | 38.4% |
| 2–4 | 79 | 38.9% |
| 5–7 | 26 | 12.8% |
| >7 | 20 | 9.9% |

*4.2. Measurement Model Test*

The measurement model test is performed to test the validity and reliability of the data that have been obtained. Validity and reliability tests were carried out by analyzing loading factors and average variance extracted, Cronbach's alpha, and composite reliability. The limit values for loading factors are 0.6 and 0.5 on average variance extracted. Meanwhile, Cronbach's alpha and composite reliability have a limit value of 0.7. Based on the measurement model test results in Table 4, it can be seen that all indicator values in each construct are above the value limit, thus fulfilling the measurement model test.

**Table 4.** Loading factors, Cronbach's alpha, composite reliability, and average variance extracted.

| Construct | Parameter | Loading Factors | Cronbach's Alpha | Composite Reliability | Average Variance Extracted |
|---|---|---|---|---|---|
| Information Experience | IE1 | 0.806 | 0.705 | 0.834 | 0.626 |
| | IE2 | 0.767 | | | |
| | IE3 | 0.799 | | | |
| Entertainment Experience | EE1 | 0.828 | 0.841 | 0.905 | 0.760 |
| | EE2 | 0.876 | | | |
| | EE3 | 0.908 | | | |
| Homophily Experience | HE1 | 0.866 | 0.855 | 0.912 | 0.775 |
| | HE2 | 0.891 | | | |
| | HE3 | 0.884 | | | |
| Relationship Experience | RE1 | 0.775 | 0.717 | 0.840 | 0.637 |
| | RE2 | 0.749 | | | |
| | RE3 | 0.865 | | | |
| Pleasure | PL1 | 0.870 | 0.857 | 0.913 | 0.778 |
| | PL2 | 0.890 | | | |
| | PL3 | 0.886 | | | |
| Arousal | AR1 | 0.887 | 0.826 | 0.896 | 0.742 |
| | AR2 | 0.825 | | | |
| | AR3 | 0.871 | | | |
| Dominance | DO1 | 0.840 | 0.896 | 0.910 | 0.717 |
| | DO2 | 0.879 | | | |
| | DO3 | 0.857 | | | |
| | DO4 | 0.810 | | | |
| Commitment | AC1 | 0.791 | 0.792 | 0.862 | 0.609 |
| | AC2 | 0.830 | | | |
| | AC3 | 0.739 | | | |
| | AC4 | 0.760 | | | |
| Intention to Follow Recommendation | PI1 | 0.834 | 0.729 | 0.843 | 0.644 |
| | PI2 | 0.855 | | | |
| | PI3 | 0.710 | | | |

Next, the discriminant validity test was conducted. This test is carried out to ascertain the correlation value in each construct. The correlation value of a construct must be higher in itself than the correlation value with other constructs. The results of the discriminant validity test are shown in Table 5.

**Table 5.** Discriminant validity test.

|  | AR | AC | DO | EE | HE | IE | PI | PL | RE |
|---|---|---|---|---|---|---|---|---|---|
| **AR** | 0.861 | | | | | | | | |
| **AC** | 0.559 | 0.781 | | | | | | | |
| **DO** | 0.542 | 0.520 | 0.847 | | | | | | |
| **EE** | 0.571 | 0.573 | 0.372 | 0.872 | | | | | |
| **HE** | 0.630 | 0.586 | 0.462 | 0.502 | 0.880 | | | | |
| **IE** | 0.501 | 0.425 | 0.543 | 0.500 | 0.515 | 0.791 | | | |
| **PI** | 0.548 | 0.599 | 0.496 | 0.475 | 0.551 | 0.494 | 0.802 | | |
| **PL** | 0.685 | 0.647 | 0.418 | 0.649 | 0.573 | 0.518 | 0.607 | 0.882 | |
| **RE** | 0.419 | 0.645 | 0.264 | 0.490 | 0.480 | 0.247 | 0.421 | 0.552 | 0.798 |

*4.3. Structural Model Test*

The structural model test was made to test the hypothesis based on the research model. The test was carried out using the partial least squares (PLS) algorithm. Testing the research model is conducted by comparing the *p*-value with a significance level of 5 percent. From the results of the structural model test in Table 6, it can be seen that there are 12 accepted hypotheses and 4 rejected hypotheses.

**Table 6.** Structural model test results.

| Hypothesis | Path | Original Sample | Sample Mean | T Statistic (∣O/STDEV∣) | *p*-Values | Result |
|---|---|---|---|---|---|---|
| H1a | Information Experience -> Pleasure | 0.192 | 0.192 | 2.786 | 0.005 ** | Accepted |
| H1b | Information Experience -> Arousal | 0.151 | 0.153 | 2.067 | 0.039 * | Accepted |
| H1c | Information Experience -> Dominance | 0.400 | 0.405 | 4.448 | 0.000 *** | Accepted |
| H2a | Entertainment Experience -> Pleasure | 0.337 | 0.337 | 4.866 | 0.000 *** | Accepted |
| H2b | Entertainment Experience -> Arousal | 0.272 | 0.270 | 3.624 | 0.000 *** | Accepted |
| H2c | Entertainment Experience -> Dominance | 0.045 | 0.041 | 0.447 | 0.655 | Rejected |
| H3a | Homophily Experience -> Pleasure | 0.185 | 0.185 | 2.515 | 0.012 * | Accepted |
| H3b | Homophily Experience -> Arousal | 0.385 | 0.383 | 5.065 | 0.000 *** | Accepted |
| H3c | Homophily Experience -> Dominance | 0.214 | 0.212 | 2.583 | 0.010 * | Accepted |
| H4a | Relationship Experience -> Pleasure | 0.251 | 0.253 | 4.709 | 0.000 *** | Accepted |
| H4b | Relationship Experience -> Arousal | 0.064 | 0.065 | 0.965 | 0.335 | Rejected |
| H4c | Relationship Experience -> Dominance | 0.040 | 0.044 | 0.504 | 0.615 | Rejected |
| H5 | Pleasure -> Commitment | 0.475 | 0.476 | 6.141 | 0.000 *** | Accepted |
| H6 | Arousal -> Commitment | 0.084 | 0.083 | 0.809 | 0.419 | Rejected |
| H7 | Dominance -> Commitment | 0.276 | 0.278 | 4.157 | 0.000 *** | Accepted |
| H8 | Commitment -> Intention to Follow Recommendation | 0.599 | 0.605 | 15.207 | 0.000 *** | Accepted |

\* *p*-value < 0.05; ** *p*-value < 0.01; *** *p*-value < 0.001.

Furthermore, the coefficient of determination was tested on the variables. The result of the test is the value of the coefficient of determination (R squared), to find out how much the ability of all the independent variables (independent) can explain the variance of the

variables that bind it (dependent). The interpretation of the test results has the criteria for R squared values of 0.67, 0.33, and 0.19 as strong, moderate, and weak [73]. The results of the coefficient of determination test can be seen in Table 7. All of them are in the moderate criteria. Based on the results of the coefficient of determination test, it can be concluded that the statistical model explains 56.3 percent of the variance of pleasure (PL) which can be predicted by information experience (IE), entertainment experience (EE), homophily experience (HE), and relationship experience (RE). In addition, 50 percent of the variance of arousal (AR) can be predicted by information experience (IE), entertainment experience (EE), homophily experience (HE), and relationship experience (RE). For the dominance (DO) variable, 34.3 percent of the variance can be predicted by information experience (IE), entertainment experience (EE), homophily experience (HE), and relationship experience (RE). Then, 49.8 percent of the variance of commitment (AC) can be predicted by pleasure (PL), arousal (AR), and dominance (DO). Finally, as much as 35.9 percent of the variance of the intention to follow recommendation (PI) can be predicted by the commitment (AC) variable. Finally, the final model of this research can be seen in Figure 2.

**Table 7.** Coefficient of determination test results (R squared).

| Variable | R Squared | R Squared Adjusted | Interpretation |
|---|---|---|---|
| Pleasure | 0.563 | 0.554 | Moderate |
| Arousal | 0.500 | 0.490 | Moderate |
| Dominance | 0.343 | 0.330 | Moderate |
| Commitment | 0.498 | 0.490 | Moderate |
| Intention to Follow Recommendation | 0.359 | 0.355 | Moderate |

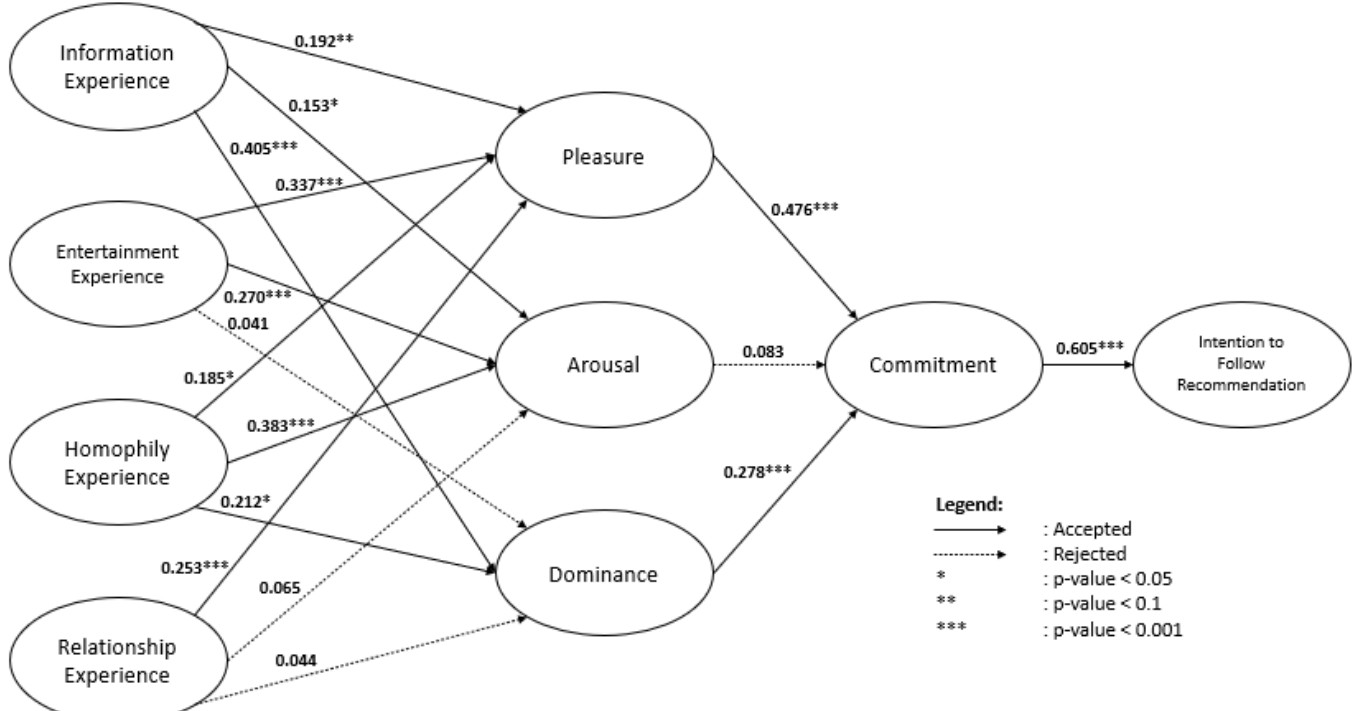

**Figure 2.** Final research model.

## 5. Discussion

Overall, this study discusses the relationship between two points, i.e., the experience of follower-influencer interactions (follower-influencer experience) and emotional dimensions. In this case, the interaction between followers and influencers on social media has a posi-

tive influence on the emotional dimensions. Then, those emotional dimensions will affect commitment, and commitment will affect aspects of intention to follow the recommendation.

In this study, there are four variables studied as follower-influencer experience points, i.e., information experience (IE), entertainment experience (EE), homophily experience (HE), and relationship experience (RE). Quoting from Maybee et al. [59], information experience is a domain that discusses how a person obtains information and meaning from information from aspects of life. In this context, information experience represents the information value of the recommendations given by influencers. Furthermore, the entertainment experience variable refers to everything related to triggering an increase in affection, physical indications, and moral motivation. In this study, this variable refers to a good experience or a sense of comfort experienced by receiving travel recommendations from influencers. Then, for the homophily experience variable, it refers to the pattern of human relationships, which tends to equate preferences or treatments with others [63]. This variable is represented by the feeling that arises from followers wanting to imitate or follow travel recommendations from influencers. Finally, according to Wang et al. [11], the relationship experience variable is one of the main factors that play a role in community commitment. In this context, relationship experience relates to the relationship and sense of belonging of the people around the followers and the followers themselves with influencers. From the results of the structural model test, it is shown that there is sufficient evidence to reject several hypotheses related to the follower-influencer experience variables. We can reject the H2c hypothesis (entertainment experience variable affects the dominance variable), the H4b hypothesis (relationship experience variable affects arousal variable), and the H4c hypothesis (the relationship experience variable affects the dominance variable). From this, it can be concluded that information experience and homophily experience have a significant influence on all three factors of pleasure, arousal, and dominance. Meanwhile, entertainment experience does not have a significant effect on dominance, and relationship experience does not have a significant effect on arousal and dominance.

Wang et al. [11] directly linked experiences with commitment, without considering pleasure, arousal, and dominance. We strengthen the model proposed by Wang et al. [11] by linking experiences with pleasure, arousal, and dominance first before commitment so that it becomes one of the contributions of our research.

In this study, there are three variables studied as points of emotional dimensions, i.e., pleasure (PL), arousal (AR), and dominance (DO). They are studied because those three variables are the three basic dimensions that indicate the state of feelings that humans have [67]. From the results of the structural model test, it is shown that there is sufficient evidence to reject the H6 hypothesis (the arousal variable affects the commitment variable). This shows that the arousal dimension does not have a significant effect on the commitment to follow the given tourist recommendations.

We can see that relationship experience does not affect arousal while at the same time it affects pleasure. Further, arousal does not affect the commitment of the influencer. We suggest that the difference between the results of the hypothesis regarding the relationship experience towards arousal and pleasure occurs because pleasure and arousal are basically different. Pleasure is about enjoyment, while arousal is about excitement. As explained in the affect grid compiled by Russell et al. [74], pleasure is in the horizontal dimension, while arousal is in the vertical dimension. Kuppens [75] also adds in his research that pleasure and arousal do not always have to be in the same direction. Low arousal may accompany the pleasant emotions of the users [75]. Thus, the follower may feel he has a relationship with the influencer, he is happy with the content of the posts from the influencer, but he does not feel enthusiastic, or his eagerness has not risen to commit to following the recommendations given by the influencer. The results of our study are almost the same as the results of research [76], that is, social interaction does not have a positive effect on arousal and in the end arousal also does not have a positive effect on satisfaction. Our study showed that relationship experience did not have a positive effect on dominance.

It means that although followers have a relationship with their influencer, however, they cannot control what travel recommendations they will receive from the influencer.

Regarding the rejection of hypothesis H2c, the results are similar to research conducted by [50]. In their research, Song et al. [50] showed that entertainment experience did not have a positive effect on dominance emotion. Their research showed that the visitor may experience enjoyment (pleasure) and excitement (arousal) from the entertainment given; however, the visitor feels that he has no control over the entertainment itself. In our research, the same thing turns out to be true in the context of travel recommendations; a travel recommendation from the influencers may entertain their followers, however, their followers may feel that they cannot control what travel recommendations they receive.

Lastly, we consider the commitment variable. Commitment is a strong and emotional relationship that a person has with something. Various studies state that commitment is one of the most important factors in building intention to follow recommendations [11]. From the results of the structural model test, it is shown that there is not enough evidence to reject the H8 hypothesis (the commitment variable affects the intention to follow the recommendation variable). This shows that the commitment aspect has a significant influence on the intention to follow the recommendation aspect.

This research contributes both in terms of practical and theoretical aspects. From a theoretical point of view, this research provides knowledge about the factors from the theory of experience that influence or do not influence followers' intention to follow travel recommendations from influencers. We have not found this specific aspect in previous studies. Then, from a practical perspective, our findings provide insight for influencers who recommend travel on their social media accounts. Based on the results of the study, the experience factors that affect the emotional dimension are information experience and homophily experience, where further, these two experience factors will influence followers to follow travel recommendations given by influencers.

There are several ways that influencers can act as one of the marketing actors who also promote tourism to their followers. The first is about the information experience experienced by followers. Information experience in the context of this research is the experience of followers related to the travel recommendation information they receive from influencers. To provide a satisfactory information experience for followers, influencers need to provide travel information in a concise and clear manner. This principle is in fact a generally accepted principle in the creation of content on social media. Social media users tend to be disinterested in content that is too long and too detailed. Some social media provides interesting features that support this. One of the most famous is the story feature of Instagram's social media. Influencers can provide short videos to provide travel recommendations with a maximum duration of 15 s. With this feature, influencers are challenged to be able to provide short travel recommendations but are still able to attract the attention of followers.

Then, the second factor, i.e., the homophily experience, is related to how far the similarities exist between followers and influencers who provide travel recommendations. The more followers feel that the influencers who provide recommendations have a lot in common with themselves, the more likely they are to follow the travel recommendations given. To increase this factor, it is very important for influencers to know the demographics and characteristics of their followers. That way, influencers can provide recommended content that is more relevant to their followers. This can be achieved by utilizing the polling feature provided on several social media channels such as Instagram and TikTok.

Influencers can also make their followers feel happy when they see content created by them. In addition, influencers also need to make it appear as if followers can control promotional content that will be carried out by influencers. Influencers can create promotional content for tourism activities that are packaged in a fun way. For example, this can be achieved by using the poll feature on social media that is submitted to followers, so that followers can choose what content the influencer needs to live with. The happy emotions expressed by influencers are expected to be accepted by their followers. Followers who

continuously feel attached or attracted to influencers have a high level of commitment. This is because of the emotion of pleasure and the feeling of dominance that are accompanied by the influence of the level of commitment of the followers. Influencers need to maintain the commitment of followers so that followers have the intention to follow the recommendations given to them.

## 6. Conclusions

This study was conducted to determine the factors of the follower-influencer interaction that affect the followers' intention to follow the travel recommendations of the influencer. From the results of the research conducted, there are four rejected hypotheses, i.e., H2c (entertainment experience variable affects dominance variable), H4b (entertainment experience variable affects arousal variable), H4c (relationship experience variable affects dominance variable), and H6 (arousal variable affects commitment variable). So, it can be concluded that the sense of belonging of followers towards influencers does not have a significant impact on the passion and dominance of the given tourist recommendations. In addition, it can also be concluded that dominance has no significant effect on the commitment to following tourism recommendations. In addition, all hypotheses can be proven in which the influential independent variable can have a significant influence on the affected variable.

The current study has several limitations in terms of respondents. The limitations are the relatively small number of respondents (203 respondents) and the geographically uneven demographics of respondents (the majority of the respondents live in the Greater Jakarta area). In the next research study, it is expected that we will be able to obtain respondents more evenly, not only in the Greater Jakarta domicile. In addition, this research has a limited scope, that is, it has not discussed or researched the content of influencers and the image brought about by influencers. For example, for an influencer who has an image or brand as an e-sports player, it is unusual to give a travel recommendation. It is necessary to investigate whether the interest felt or obtained by followers will be the same as for the travel recommendations given by influencers who have the image of traveling experts. Another example is the need to research whether the interest felt by followers when receiving travel recommendations in the form of Instagram Stories will be the same as the interest obtained when receiving travel recommendations in the form of threads on Twitter. For further research, it can be considered that this research can open new discussion avenues regarding whether the type of content provided by influencers affects the interest in travel recommendations for followers and whether the type of influencer affects followers' interest in the travel recommendations provided.

**Author Contributions:** Conceptualization, B.P., A.R. and A.N.H.; methodology, B.P., A.R. and A.N.H.; software, A.F.H., F.H.F., N.H.W. and R.H.S.; validation, A.R, A.N.H. and K.P.; formal analysis, A.N.H., A.F.H., F.H.F., N.H.W. and R.H.S.; investigation, A.R., A.N.H., A.F.H., F.H.F., N.H.W. and R.H.S.; resources, A.F.H., F.H.F., N.H.W. and R.H.S.; data curation, A.F.H., F.H.F., N.H.W. and R.H.S.; writing—original draft preparation, A.R., A.F.H., F.H.F., N.H.W. and R.H.S.; writing—review and editing, B.P., A.R. and A.N.H.; visualization, A.R., A.F.H., F.H.F., N.H.W. and R.H.S.; supervision, B.P., A.R., A.N.H. and K.P.; project administration, B.P.; funding acquisition, B.P. All authors have read and agreed to the published version of the manuscript.

**Funding:** PUTI Grant contract no. NKB-4371/UN2.RST/HKP.05.00/2020 funded by Universitas Indonesia.

**Data Availability Statement:** Not applicable.

**Conflicts of Interest:** The authors declare no conflict of interest.

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
