# Peer review of "The Effect of Interaction between Followers and Influencers on Intention to Follow Travel Recommendations from Influencers in Indonesia Based on Follower-Influencer Experience and Emotional Dimension"

_information, doi:10.3390/info13080384_

Round 1

Reviewer 1 Report

The paper studies the effect of interaction between followers and influencers by means of SEM model towards the use of follower-influencer experience and the emotional dimensions theories.

1. There are several typo errors in the paper such as:
- the "often" in line 40,
- the phrase "is affect" in line 74,
- legend in Figure 2 is still in Bahasa/Indonesian, etc.
Double check entire papers.

2. The fact that some hypothesis are rejected, I tempted to think that the overall model might be inconsistent or might be due to some confusing statements in the questionnaire that misled the answer. Although the writer had proved that they statistically are valid, the writer should provide clear explanation to the several rejection hypothesis, such as why does the relationship experience not affect the arousal while at the same time the relationship experience does affect the pleasure? The term pleasure and arousal are somehow quite difficult to differentiate, aren't they?  Also other rejection i.e. why does the arousal not affect the commitment while the pleasure does affect the commitment? Is the model in the study i.e. follower-influencer experience and the emotional dimensions affect the commitment valid?

3. The writer should provide explanation to different finding between this paper and the existing findings in Wang et al., [11] which stated that information experience does not have a significant effect on pleasure, arousal, and dominance, while the findings in this paper is completely the other way around.

Reviewer 2 Report

The concept of social media influencers' effect is very up-to-date and important especially in the context of tourism. However, authors have mostly left out the tourism component in their literature review. All used constructs must be connected to the tourism setting, were there any previous relevant studies, why did you choose PAD model specifically? Was it related to tourist behavior in the previous literature? A lot of the paper is repetitive, authors have repeated definitions of the constructs several times which is not necessary in my opinion. As for the influencers, authors have neglected to ask the respondents who are their influencers, who are they following. How can you be sure that those they consider influencers are even relevant to the travel related content? Authors did not explain who they specifically consider as influencers - how many followers do they have, how long are they active on social media, how often do they post, etc. I assume that the scales were translated but authors do not mention it anywhere. What was the previous reliability of the used scales? Finally, authors do not explain why there was no influence of certain constructs, why are their findings different, what could be the possible explanation for their results. I hope my comments are clear and helpful. Best of luck!

Round 2

Reviewer 2 Report

Thank you for your effort, the paper seems much improved. I would just add that better control of who is considered an influencer should be included in the future research and can be viewed as a limitation of this one.